# Preparation and Stabilization of High Molecular Weight Poly (acrylonitrile-*co*-2-methylenesuccinamic acid) for Carbon Fiber Precursor

**DOI:** 10.3390/polym13223862

**Published:** 2021-11-09

**Authors:** Shuxian Zhang, Yanjin Dang, Xuepeng Ni, Chunshun Yuan, Huifang Chen, Anqi Ju

**Affiliations:** 1Shanghai Key Laboratory of Lightweight Composite, College of Materials Science and Engineering, Donghua University, Shanghai 201620, China; cameron68@163.com (S.Z.); 13918539856@163.com (Y.D.); nxp0127@163.com (X.N.); yuanchunshun@126.com (C.Y.); hfchen@dhu.edu.cn (H.C.); 2State Key Laboratory for Modification of Chemical Fibers and Polymer Materials, Donghua University, Shanghai 201620, China; 3Key Laboratory of High-Performance Fibers & Products, Ministry of Education, Donghua University, Shanghai 201620, China

**Keywords:** 2-methylenesuccinamic acid, mixed solvents polymerization, carbon fiber stabilization, thermodynamic properties

## Abstract

Bifunctional comonomer 2-methylenesuccinamic acid (MLA) was designed and synthesized to prepare acrylonitrile copolymer P (AN-*co*-MLA) using mixed solvent polymerization as a carbon fiber precursor. The effect of monomer feed ratios on the structure and stabilization were characterized by elemental analysis (EA), Fourier transform infrared spectroscopy (FTIR), gel permeation chromatography (GPC), X-ray diffraction (XRD), proton nuclear magnetic (^1^H NMR), and differential scanning calorimetry (DSC) for the P (AN-*co*-MLA) copolymers. The results indicated that both the conversion and molecular weight of polymerization reduce gradually when the MLA content is increased in the feed and that bifunctional comonomer MLA possesses a larger reactivity ratio than acrylonitrile (AN). P (AN-*co*-MLA) shows improved stabilization compared to the PAN homopolymer and poly (acrylonitrile-acrylic acid-methacrylic acid) [P (AN-AA-MA)], showing features such as lower initiation temperature, smaller cyclic activation energy, wider exothermic peak, and a larger stabilization degree, which are due to the ionic cyclization reaction initiated by MLA, confirming that the as-prepared P (AN-*co*-MLA) is the potential precursor for high-performance carbon fiber.

## 1. Introduction

Owing to their unique performance, such as high strength, high temperature resistance, good flexibility, and lightweight, carbon fibers are widely used in aerospace and other high-tech equipment. Polyacrylonitrile- (PAN) based carbon fiber occupies 90% of the market [1,2,3,4], and the quality of the PAN precursor results in the final performance of the carbon fiber. However, the polarity between the cyano groups of the acrylonitrile homopolymer usually causes the gelation of the spinning solution and results in poor spinnability [5]. More importantly, the polymerization of the PAN homopolymer is only triggered through a radical mechanism that leads to a high cyclization temperature, centralized heat release, and poor carbon fiber performance [6,7]. To solve these problems, a small quantity of comonomers is usually introduced to the PAN polymer chains to reduce the intermolecular force and to improve the stabilization. The most used monomers are acidic comonomers, such as itaconic acid (IA) [8], and neutral comonomers, such as as methyl acrylate (MA) [9,10] and methacrylate (MMA) [11]. IA can promote the stabilization through ionic cyclization reactions with a lower initiation temperature and with a smaller activation energy due to the fact that IA contains two carboxyl groups. However, in the polymerization process, the carboxyl groups in IA can trap free radicals and then bind to them and release protons (H^+^), which is not conducive to the improvement of the molecular weight of the polymer, and it has a significant effect on promoting the tensile strength of the final carbon fiber. In recent years, the acrylamide (AM) [12,13,14] has attracted the attention of researchers due to its unique properties. The nitrogen atom of the amide group can carry out a nucleophilic attack on the adjacent carbon atoms of the nitrile group, which can improve the solubility of PAN and can initiate the cyclization reaction of the nitrile group through an ion mechanism. There is no H^+^ release in the process of polymerization, which is beneficial to improving the molecular weight of the PAN copolymer.

For the above reasons, in this work, a bifunctional comonomer 2-methylenesuccinamic acid (MLA) containing carboxyl and amide groups was synthesized and used as a comonomer to prepare the poly (acrylonitrile-*co*-2-methylenesuccinamic acid) [P (AN-*co*-MLA)] binary copolymer using mixed solvent method. The difference between MLA and IA is that the amide group in MLA can replace an IA carboxyl group, which can lower the chain termination occurrence that takes place during the polymerization process, thus helping to increase the molecular weight of the final PAN polymer. MLA also has great benefits in improving the performance of carbon fiber. At the same time, we know that the radicals can transfer to dimethyl sulfoxide (DMSO) in solution polymerization [15,16,17]. However, deionized water has a very small chain transfer constant that is close to 0, which can avert the occurrence of the chain transfer and can improve the molecular weight of the PAN polymer. Therefore, a mixed solution of dimethyl sulfoxide/deionized water = 60/40 (wt/wt) was adopted as the reaction media to prepare P (AN-*co*-MLA) in this work.

P (AN-*co*-MLA) with different compositions was acquired by changing the mass feed ratio of AN/MLA. The effect of the monomer feed ratios on polymerization, polymer structure, and stabilization were studied by X-ray diffraction (XRD), elemental analysis (EA), Fourier transform infrared spectroscopy (FTIR), and differential scanning calorimetry (DSC). The results prove that P (AN-*co*-MLA) has a molecular weight that is as high as 590 kg/mol, which is much higher than the molecular weight of that prepared by solution polymerization of 160 kg/mol [18]. Compared to the PAN homopolymer, the stabilization of P (AN-*co*-MLA) was notably improved with a lower initiation temperature, smaller cyclic activation energy, and a higher stabilization degree due to the ionic cyclization reaction that was initiated by MLA.

## 2. Experimental

### 2.1. Materials

Acrylonitrile (AN, analytically pure) was bought from Shanghai East China Reagent Supply and Marketing Co., Ltd. (Shanghai, China) and was distilled at regular pressure to collect a 77–78 °C fraction. The initiator azodiisobutyronitrile (AIBN, analytically pure) was provided by Shanghai Macklin Biochemical Co., Ltd. (Shanghai, China) and was recrystallized with methanol. The compound 2-Methylenesuccinamic acid (MLA) was made in the laboratory. Itaconic acid (IA), dimethyl sulfoxide (DMSO, analytically pure), and ammonia hydroxide (28 wt%, analytically pure) were all purchased Sinopharm Group Chemical Reagent Company (Shanghai, China).

### 2.2. Synthesis of MLA

An amount of 154 g of ammonia hydroxide (28%) was added to a 1 L round bottom flask and was cooled to 0 °C. An amount of 98.0 g of itaconic acid was slowly added to the round bottom flask while maintaining the reaction temperature in the flask below 5 °C. Then, the achieved mixture was mechanically stirred at room temperature for 2 h after the itaconic acid was completely added to the solution. When the temperature of the mixed solution dropped to 0 °C, 4 mol/L diluted sulfuric acid drops were added until the pH value of the solution was less than 1. Then, the resultant emulsion was continually stirred at room temperature for 18 h. After that, the suspension was filtered and washed with 100 mL diluted sulfuric acid (PH < 3). Finally, the collected MLA was recrystallized with absolute ethanol (120 mL) to obtain a purified product for the next step experiment. The MLA synthesis process is shown in Appendix A.

Crystal, 52.7% yield. IR (KBr) *v*_max_ cm^−1^: (C–O) 1282, 1170, (δ  C–H) 1433, (C=C) 1640, 1668, 1704 (C=O), (N–H_2_) 3359, 3200, (C–H) 2934, 2901. ^1^H-NMR (TMS, RT, DMSO-d_6_, 400 Hz) δ ppm: 12.39 (s, 1H, COOH), 7.26, 6.79 (s, 2H, H_2_N), 6.09 (d, *J* = 0 Hz, 1H, CH_2_=), 5.64 (d, *J* =0 Hz, 1H, CH_2_=), 3.07 (s, 2H, CH_2_) [18], Appendix A.

### 2.3. Preparation of P (AN-co-MLA) in Mixed Solvents

The copolymerization was prepared in a three-neck round-bottom flask at 60 °C with a nitrogen atmosphere by mixing solvents of DMSO/deionized water with a mass ratio = 60/40 for the reaction medium. First, AIBN (0.49 g), MLA (0.33 g), DMSO (60 g), and H_2_O (40 g) were added into the three-neck round-bottom flask, which was then degassed and filled with nitrogen. Next, AN (32.67 g) was added into the flask through the upper end of the condenser tube and was heated up to 60 °C with medium-speed stirring (120 r/min), and the stirring speed was reduced (90 r/min) after 6 h. The polymerization solution was continuously conducted for 24 h at 60 °C. The resultant emulsion was filtered and washed with methanol several times, and the resultant P (AN-*co*-MLA) was dried at 75 °C in a vacuum dryer overnight. (Figure 1). To study the effect of monomer ratios on the P (AN-*co*-MLA), the changes in the mass feed ratio of the MLA in terms of the total monomers varied 1 to 5%, with the total mass of AN and MLA remaining unchanged. In addition, the PAN homopolymer and terpolymer acrylonitrile-acrylic acid-methyl acrylate P (AN-AA-MA) with mass ratio 98/2/2 were also prepared for comparison.

### 2.4. Polymer Characterization

The average molecular weight of the viscosity of acrylonitrile was measured using the Ubbelohde viscometer method in a (50 ± 0.5) °C water bath. The calculation formula was as follows: $\left[\eta \right]=2.83*10^{-4}M_{\nu }^{0.759}$: *η* represents the intrinsic viscosity calculated through linear extrapolation in Appendix A [19]. The copolymer compositions were measured using a Nicolet 8700 FTIR spectrophotometer at room temperature. An amount of 200 mg of potassium bromide and 1 mg of sample were grinded together evenly and were pressed into a sheet. The samples were determined using 32 scans at a 1 cm^−1^ resolution. Gel permeation chromatography (GPC) was performed on a PL-GPC 50 integrated GPC system with a refractive index detector using dimethylacetamide as the mobile phase at 25 °C. The proton nuclear magnetic ^1^H NMR (400 MHZ) was recorded using DMSO-d6 as a solvent on a Bruker DMX-400 spectrometer to analyze the molecular chain structure of the copolymer. The thermal behavior was carried out on a DSC 822. For the DSC characterizations, the samples were heated at different heat rates at a range between 50 °C and 375 °C under a N_2_ atmosphere (40 mL/min). X-ray diffraction (XRD) patterns were measured to determine the crystallite size (*Lc*) and the crystallinity of P (AN-*co*-MLA) at a speed of 3°/min at temperatures between 5–60° on a D2 phaser. All of the XRD tests were in the same conditions. The average crystallite size (*Lc*) and interplanar spacings (*d*) were determined using the Scherrer equation and the Bragg equation, respectively [20]: $d=\frac{\lambda }{2sin\theta },\;Lc=k\lambda /\beta cos\theta$, where *k* is a constant 0.89, β is the full width at half maximum intensity around *2*θ=17*,* and θ is the Bragg angle. Meanwhile, the element content of the P (AN-*co*-MLA) copolymers including carbon (C), hydrogen (H), nitrogen (N), and oxygen (O) were determined using an Elementar Vario El III elemental instrument analyzer.

To further explore the thermal stability of P (AN-*co*-MLA), the samples were heated to 200 °C and were for 30 min at a rate of 1 °C/min under an air atmosphere in a tubular furnace.

## 3. Results and Discussion

### 3.1. Reactivity Ratio Studies

The molecular weights of MLA and AN are 129 g/mol and 53 g/mol, respectively. According to the mechanism of free radical polymerization, ignoring the terminal group effect of the polymer, the oxygen in the P (AN-*co*-MLA) only exists in MLA, so we can calculate the molar fraction of MLA in P (AN*-co*-MLA) according to the following formula:(1)$$\left[MLA\right]=\frac{C_{0}/\left(3*16\right)}{\frac{1-\left(C_{0}*129\right)/\left(3*16\right)}{53}+\frac{C_{0}}{3*16}}$$
where $C_{0}$ is the oxygen content of the P (AN-*co*-MLA) copolymers calculated by element analysis.

The Fineman–Ross and Kelen–Tüdõs methods can be used to calculate the monomer reactivity ratios. According to the Mayo–Lewis equation, the Fineman–Ross method can be expressed as follows [21]:(2)$$G=r_{1}H-r_{2}$$

*G* and *H* can be gained from the following equations:(3)$$G=\frac{X\left(Y-1\right)}{Y},\;H=\frac{X^{2}}{Y}$$

*X* and *Y* are the molar fraction ratios of AN/MLA in the total monomer and P (AN-*co*-MLA) copolymers, respectively:(4)$$X=\frac{\left[M_{1}\right]}{\left[M_{2}\right]},\;Y=\frac{d\left[M_{1}\right]}{d\left[M_{2}\right]}$$

There is a linear relationship between *G* and *H*, where the slope is $r_{1}$, and an intercept is $r_{2}$.

The linear relationship equation proposed by Kelen–Tüdõs is as follows [22]:(5)$$\eta =\left(r_{1}+\frac{r_{2}}{\alpha }\right)\xi -\frac{r_{2}}{\alpha }$$

η and ξ are calculated by
(6)$$\eta =\frac{G}{\left(\alpha +H\right)},\;\xi =\frac{H}{\left(\alpha +H\right)}$$

α is a constant and is represented by
(7)$$\alpha ={\left(H_{m}H_{M}\right)}^{-\frac{1}{2}}$$
where $H_{m}$ and $H_{M}$ are the minimum and maximum values in a series of measurements. The relationship between *η* and *ξ* is a straight line, extrapolating the line to *ξ* = 1 yields *r*_1,_ and extrapolating it to *r*_2_*/α* = 0 obtains *r*_2_. The parameters of the P (AN-*co*-MLA) polymers prepared at a low conversion are shown in Table 1, and the monomer reactivity ratios are evaluated by the Fineman–Ross and Kelen–Tüdõs method and are shown in Figure 1.

It can be seen from Figure 1 that the monomer reactivity ratio results calculated by the above two methods are very similar, and the reactivity ratios of the MLA monomer is higher than that of the AN monomer. Therefore, the free radical is biased towards the comonomer MLA, which means that MLA is more likely to enter the polymer chain than acrylonitrile is. This result is further confirmed again by elemental analysis (*Ea*).

### 3.2. Molecular Weight and Elemental Analysis Studies

The effects of the monomer feed ratios (AN/MLA) on the molecular weight and the polymerization conversion are shown in Figure 2. It can be seen that a high polymer conversion (75%) and high molecular weight (590 kg/mol) are obtained when the monomer feed ratio is 99/1. Both of them gradually decrease as the amount of MLA increases in the feed. This is because MLA possesses a larger molecular size and more steric hindrance than AN. The introduction of the comonomer MLA hinders the growth of chain free radicals, which reduces the polymerization conversion and affects the increase of the molecular weight of the P (AN-*co*-MLA) copolymers. Therefore, the polymerization conversion rate is less than 50% when the mass ratio of AN/MLA is 96/4. In addition, the results show that the polymer dispersity index (PDI) of the P (AN-co-MLA) copolymers are between 1.58 and 2.75 in Table 2 and in the GPC curves (Appendix A). In the work, the polymerization media was mixed solvent DMSO/water in this work, and the chain transfer constant of water is 0 [23]; therefore, the chain transfer constant of AN radicals to the solvent is reduced, and the PDI of P (AN-co-MLA) is narrow. However, the value of PDI gradually increased with the increase of MLA, which may be attributed to the large steric hindrance and high reactive ratio of MLA. It is generally believed that narrow PDI and high molecular weight are better for high performance carbon fiber [24]. Hence, considering the actual industrial applications, the feed ratio of MLA is better when it is less than 3%.

Figure 3 provides the elemental analysis of the P (AN-*co*-MLA) copolymers with diverse mass feed ratios of AN/MLA. The only source of O content is from the comonomer MLA in the P (AN-*co*-MLA) copolymers. It can be seen that the O content in P (AN-*co*-MLA) gradually increases when the amount of MLA increases in the feed, while the contents of C, N, and H decrease slightly. At the same time, the O content in the feed (O_f_) is also provided in Figure 3. Obviously, the O content in the P (AN-*co*-MLA) copolymers is higher than the content of O in the feed, confirming that more MLA entered the polymer chains than the feed, which echoes the reactivity ratio results above.

### 3.3. FTIR and NMR Studies

In order to confirm the composition of PAN copolymers, the different samples were determined using ^1^H spectroscopy in Appendix A and Figure 4 [25]. The signal at δ=2.05 ppm can be ascribed to the —CH_2_ protons of AN and MLA [24]. The chemical shifts at δ=3.09 ppm are assigned to the —CH proton in AN. Additionally, signals can be observed at 6.97 ppm for —NH_2_ and at 10.95 ppm for —COOH. The FITR spectra in Figure 5a shows the different feed ratios of the AN/MLA of the P (AN-*co*-MLA) and PAN homopolymer. The infrared spectrum of the P (AN-*co*-MLA) copolymer shows a C=O stretching vibration peak and an amide II band stretching vibration peak at 1718 cm^−1^ and 1678 cm^−1^, respectively. Compared to the PAN copolymers, there are not any stretching vibration peaks at 1718 cm^−1^ and 1678 cm^−1^ in the PAN homopolymer, and since only comonomer MLA has the C=O and amide groups, the FTIR result indicates that the MLA copolymerizes with AN successfully. Additionally, as the proportion of the MLA amount increases in the feed, the C=O absorption peaks intensity become stronger, suggesting the MLA content increases in the copolymer, which is in a good agreement with the above elemental analysis results. Therefore, the structural composition of the P (AN-*co*-MLA) copolymer can be adjusted by the monomer feed proportion of AN/MLA. The bands at 2244 cm^−1^ that have been assigned to the stretching vibration peak of C≡N show no significant changes in the intensity for all of the P (AN-*co*-MLA) copolymers. This indicates that the AN units exhibit long chain continuity in the acrylonitrile copolymer. At the same time, the in-plane bending vibration peaks of C-H in CH, CH_2_, and CH_3_ at 2939 cm^−1^ and of C-H in -CH_2_ at 1454 cm^−1^ remain unchanged in all of the polymers.

The FITR spectra of the P (AN-*co*-MLA) stabilized at 200 °C in air for 30 min are shown in Figure 5b. There is an apparent change for the spectra at 2244 cm^−1^ that belongs to the stretching vibration peak of C≡N. As the proportion of MLA feed in the total amount decreases, the stretching vibration of C≡N becomes weak. The reason for this is that the C≡N structure converts into C=N during the cyclization reaction in the heat treatment process, and the linear polymer turns into a cyclic polymer as well [26,27]. The stretching vibration peaks at 1718 cm^−1^ disappear; meanwhile, new broad peaks at 1607 cm^−1^ emerge in Figure 5b, which belongs to the stretching vibration of C=N (cyclization reaction) conjugated with C=C (dehydrogenation reaction). At the same time, with the increase of the comonomer content, the absorption peak at 1607 cm^−1^ becomes stronger, indicating that the introduction of the comonomer can improve its cyclization and dehydrogenation [28]. The intensity of the absorption peaks at 2244 cm^−1^ and 1607 cm^−1^ were used to assess the degree of stabilization. To evaluate the degree of stabilization, a parameter (*Es*) is defined, and the equation is shown below:(8)$$Es=\frac{A_{1607cm^{-1}}}{A_{2244cm^{-1}}}$$

*A = log(T_0_/T)* is defined as the absorbance intensity, *T*_0_ is the transmittances at baseline, and *T* is the transmittances at maximum. In the Lambert–Beer law: *A* = *abc*, where a represents the molar absorption coefficient (mol cm^−1^), *b* represents the slice thickness of the KBr pellet (cm), and *c* is the molar concentration of sample (mol cm^−1^). The peak intensities at 2244 cm^−1^ and 1607 cm^−1^ are measured by employing the same KBr sample slice, which means that the *b* values are the same. Thus, the intensity ratio of the peaks at 1607 cm^−1^ and 2244 cm^−1^ is approximately equivalent to the ratio of the amount of the generated C=N and C=C in the residual C≡N group, and the *Es* can be used to reflect the extent of stabilization.

As illustrated Figure 6, the *Es* of P (AN-*co*-MLA) continuously increase when the MLA content in the feed increase, which is higher than the stabilization seen in the PAN homopolymer, confirming that the incorporation of MLA into the PAN chains can improve the stabilization of PAN effectively, which the more MLA being present, the better. As we all know, the degree of stabilization is significant for the performance of the final resulting carbon fiber, and it is better to process the stabilization degree as fully as possible in order to acquire an ideal carbon fiber during stabilization reactions. It is generally believed that the cyclization reaction can only be initiated under high temperatures by a free radical mechanism reaction in the PAN homopolymer, as shown in Figure 2, which results in a concentrated heat release in a short time and that is finally able to break the molecular chain of the PAN homopolymer, while the cyclization reaction can be initiated through two kinds of mechanisms, including the ionic mechanism and the free radical mechanism for the P (AN-*co*-MLA) copolymers, as shown in Figure 3. In one of these mechanisms, the carboxyl oxygen of MLA can create a nucleophilic attack on the carbon atom of the adjacent nitrile group. For the other mechanism, the amide nitrogen of MLA can also create a nucleophilic attack on the nearby carbon atom, which, as seen in Figure 3, can be shown to induce cyclization.

To explore the impact of temperature on stabilization, the P (AN-*co*-MLA) copolymer with a mass feed ratio AN/MLA = 97/3 is heated at various temperatures for 30 min, and the FTIR spectrum is shown in Figure 7. The intensity of the C≡N stretching vibration peak of the P (AN-*co*-MLA) copolymer at 2244 cm^−1^ gradually declines with the increase of the stabilization temperature from 160 °C to 280 °C, showing that the groups have changed from C≡N to C=N [28]. However, the intensity of the conjugate stretching vibration peaks of -C=C- and C=N at 1607 cm^−1^ increase gradually before 200 °C. After 200 °C, the change of the 1607 cm^−1^ peak is particularly obvious, indicating that the cyclization and dehydrogenation reactions are accelerated when the temperature is higher than 200 °C, and more C≡N turns into C=N [29] The calculated *Es* of the P (AN-*co*-MLA) copolymers with feed ratios AN/MLA = 97/3 (wt/wt) stabilized at different temperatures (160 °C–280 °C) for 30 min and are shown in Figure 8. This shows that there is only a slight change in the *Es* under 200 °C, so only a small number of cyclization reactions occurs. Above 200 °C, the changes in the *Es* happen quickly, indicating that a mass of dehydrogenation, and it becomes clear that cyclization reactions take place in this temperature range. Finally, the changes in the *Es* slow down at 280 °C, indicating that most of the dehydrogenation and cyclization reactions have already taken place within temperature range of 200 °C–260 °C. Similar to the P (AN*-co-*MHI) copolymer, the nitrogen nucleophilic in the amide group in the MLA attacks the carbon atoms of the adjacent cyanide groups to initiate cyclization, which promotes the stabilization of P (AN*-co-*MLA) [30,31]. In short, the treatment temperature has a great effect on the stabilization of P (AN-*co*-MLA), which should be well controlled in the practical production of carbon fibers.

### 3.4. XRD Studies

The copolymers with various mass ratios of AN/MLA and PAN homopolymer were investigated by XRD analysis in Figure 9a. This analysis shows that the peak intensity of the PAN homopolymer is higher than that of the other copolymers, and the CI value is larger because there is no comonomer that has been introduced, so the crystal structure of the homopolymer is not destroyed. Additionally, two diffraction peaks appear in the XRD curves: a sharp peak at 2θ=17° that can be ascribed to (100) the orthorhombic structure of the PAN crystals and a weak peak at 2θ=29° that belongs to (101) the crystalline plane of the hexagonal lattice of PAN [32,33]. XRD patterns were processed using the origin 9.5 software to analyze the peak center and the full width at half maximum (FWHM) at around 2θ=17°. In view of the above results, the following data can be calculated: crystallite size *Lc,* the matched interplanar spacing *d,* and crystallinity index *CI,* which are tabulated in Table 3. As shown in Table 3, the *d* values do not show obvious changes when the MLA content in the commoner increase because the peaks at around 2θ=17° have almost no fluctuations. However, *CI* shows a declining trend as the MLA content increases in the feed. The reason for this is that the incorporation of MLA into the PAN chains can block the interaction among the intermolecular C≡N structure.

Figure 9b exhibits the XRD spectra of P (AN-*co*-MLA) copolymers heated at 200 °C for 30 min under an air atmosphere. As we know that stabilization is the structural change process that allows the structure to change from a linear structure to a ladder structure, this coincides with the peak intensity changes at around 2θ=17°. Hence, the peak intensity changes can be used to assess the degree of stabilization according to Figure 9a,b, and *SI* is presented in the equation below [25,27,28]:(9)$$SI=\left(I_{0}-I_{S}\right)/I_{0}$$
where $I_{S}$ represents the strength of peak at around 2θ=17° after being heated 200 °C for 30 min, and $I_{0}\;$ is the peak intensity at around 2θ=17° of the original P (AN-*co*-MLA). Based on the results in Table 4, the *SI* value of the PAN homopolymer is negative because the stabilization has not started. The *SI* is also negative when the AN/MLA feed ratio is 99/1, not much cyclization that was initiated through the ionic mechanism, and there was no obvious stabilization reaction of the P (AN*-co-*MLA) copolymer which can be attributed to the small amount of MLA in the polymer chains. However, the heating energy at 200 °C is enough to break the boundaries between the crystalline and amorphous zones, resulting in further crystallization [34]. The *SI* becomes positive when the AN/MLA feed ratio is 98/2, and *SI* shows an increasing trend as the MLA content increases. The *SI* value (34.66%) of AN/MLA = 95/5 is the largest one among the P (AN*-co*-MLA) copolymers, where different monomer mass ratios are demonstrated. This shows that a great deal of MLA was incorporated into the polymer when the proportion of MLA content in the total feed increased, thus initiating more cyclization through the ionic mechanism by MLA at around 180 °C (Table 5).

The XRD spectra of the P (AN-*co*-MLA) with AN/MLA feed ratio = 97/3 under different stabilization temperatures from 160 °C to 280 °C for 30 min are displayed in Figure 10. The *SI* values in Table 5 were calculated according to the results in Figure 10. The strength of diffraction peak ( 2θ=17°) becomes weaker as the temperature increases, especially when the temperature is above 200 °C. The characteristic diffraction peak at 2θ=29° gradually disappears at temperatures higher than 200 °C. In the meantime, a new diffraction intensity peak appears at 2θ=25° because of the ladder structure. The above phenomena confirms that the polymer chains of P (AN-*co*-MLA) gradually transfer from the linear structure to the ladder structure when the temperature increases. In Table 5, the value of the *SI* is a negative number at 160 °C, and the stabilization process has not yet started. The heating energy can cause P (AN-*co*-MLA) to further crystallize and results in a negative *SI* value. In addition, when the temperature increases, the *SI* value changes from negative to positive, and the value increases very quickly at 200 and 220 °C and then changes slowly above 240 °C, which agrees well with the FITR results.

### 3.5. DSC Analysis

DSC is the most widely used thermal analysis that is used to study the exothermic phenomena of the cyclization reaction of PAN [35]. PAN has poor thermal conductivity, so a small amount of comonomer is usually added. Han et al. copolymerized acrylonitrile with vinyl acetate to form a copolymer, and the comonomer could be used as a defect in the copolymer to improve thermodynamic performance, which is conducive to reducing the dipole–dipole interaction in the PAN system, reducing the stabilization temperature [19,36]. The DSC diagram of the PAN homopolymer, the P (AN-AA-MA) terpolymer, and the P (AN-*co*-MLA) copolymer were performed under a nitrogen atmosphere at a rate of 10 °C/min from room temperature to 350 °C in Figure 11. The parameters that are available in the figure include the initiation temperature ($T_{i}$), termination temperature ($T_{f})$ and their difference ($\Delta T=T_{f}-T_{i}$), the first peak temperature ($T_{P1}$, the peak at the lower temperature), the second peak temperatures ($T_{p2}$, peaks at higher temperatures), the third peak temperatures ($T_{p3}$, peaks at higher temperatures), the reaction heat (ΔH), and the rate of heat release (ΔH/ΔT) [37,38], and these parameters are summarized in Table 6.

A DSC test was performed under a nitrogen atmosphere, where only cyclization reactions occur. Under these conditions, it is possible for a sharp exothermic peak to occur in the PAN homopolymer, and a high cyclization initiation temperature of 244 °C is required, as depicted in Figure 11. The main reason for this is that the cyclization of the PAN homopolymer only can be initiated by the free radical mechanism [39,40,41]. In the meantime, a lot of heat would be released during this process, causing the molecular chain to break. [42] Additionally, the molecular chain will eventually inherit these defects into the resulting carbon fiber. In spite of the initiation temperature being lower to 203 °C for P (AN-MA-AA) terpolymer, there is still only one exothermic peak in the DSC curves. For the P (AN-*co*-MLA) copolymers, two peaks or even three exothermic peaks appear when the feed ratio is ≤97/3, which broadens the exothermic peak and avoids concentrated exothermic reactions. The new exothermic peak occurs at lower temperatures because the cyclization of P (AN-*co*-MLA) can be initiated through an ionic mechanism. Both the oxygen of carboxyl group and the nitrogen in the amide group of MLA can attack the carbon atom of the adjacent nitrile group to trigger the cyclization reactions. Toms et al. determined when the feed ratios of AN/AA increase, the ion mechanism dominates the cyclization reaction, but as the introduction rate of the AA comonomer into polymerization decreases, it is initiated by an ionic and radical mechanism because its more uniform distribution in the chain leads to analogous effects and results in a decrease in the cyclization activation energy. [43] In this work, the lowest exothermic peak is due to the cyclization reaction created by oxygen attack, the middle one is triggered by amide nitrogen, while the highest one is caused through radical cyclization. These three cyclization reactions are a mutual competition relationship, and the free radical mechanism plays a dominant role when the AN/MLA feed ratio is 99/1 and 98/2; instead, the amide nitrogen attack dominates the cyclization when the feed ratio is 96/4 and 95/5. As displayed in Table 6, the initiation temperature $T_{i}\;$ of the P (AN-*co*-MLA) copolymer is less than that of the PAN homopolymer and P (AN-MA-AA), showing a decreasing tendency when the MLA amount is augmented. This decrease indicates that the cyclization reaction of the P (AN-*co*-MLA) copolymer is easier to initiate using the ionic mechanism and that it is easier for the stabilization reaction to proceed at lower temperatures compared to the PAN homopolymer [24]. Additionally, the *ΔH* value of P (AN-*co*-MLA) gradually increases as the mass ratio of the MLA content increases in the feed, implying more cyclization reactions. Simultaneously, the introduction of the comonomer MLA can relax the exotherms in the process of stabilization with smaller ΔH/ΔT and can slow the rate of heat release, which is good for the preparation of high-performance carbon fibers [44].

### 3.6. Evaluation of Activation Energy (Ea) of Cyclization Reactions

Evaluation of the activation energy for cyclization reactions can be further carried out based on the DSC curves of the PAN homopolymer, P (AN-AA-MA) = 98/2/2, and P (AN-*co*-MLA) copolymer, with a mass feed ratio AN/MLA = 97/3 heated from 50 °C to 400 °C under the N_2_ atmosphere at various rates (5, 10, 15, 20, 30 °C/min) in Figure 12.

Overall, the exothermic peaks move to high temperatures as the heating rate increases for all of the PAN polymers. The activation energy *(Ea)* can be calculated using Kissinger and Ozawa’s method because this method only needs a sequence of DSC curves that have been heated at diverse rates to deduce the *Ea* without any knowledge of the reaction mechanism.

The *Ea* was determined by the following equations in Kissinger’s method [45]:(10)$$-\frac{E_{a}}{R}=\frac{d\left[\ln(\varphi /T_{m}{}^{2}\right]}{d\left(1/T_{m}\right)}$$

*Ea* was calculated from the slope of the linear plot of $\ln(\varphi /T_{m}{}^{2})$ versus 1000/$T_{m}$, as shown in Figure 13a.

Ozawa’s method uses the following equation [46]:(11)$$-\frac{E_{a}}{R}=2.15\frac{d(\log\varphi )}{d\left(1/T_{m}\right)}$$
where $T_{m}$ is the maximum temperature in the DSC curves, and φ represents the heating rate, R = 8.314. *Ea* was calculated from the slope of the linear plot of logφ versus 1000/$T_{m}$, as shown in Figure 13b.

The *Ea* values that were calculated by the above two methods are listed in Table 7. The figure shows that the *Ea* value calculated by these two methods agrees well for the PAN homopolymers or all of the PAN copolymers, and it can be seen that the *Ea* values of the PAN homopolymer and the P (AN-MA-AA) copolymer were approximately 168 kJ/mol and 140 kJ/mol, respectively, during this cyclization reactions. Compared to the PAN homopolymer and the P (AN-MA-AA) copolymer, the activation energy of the P (AN-*co*-MLA) copolymer was divided into two or three parts and was lowered by the introduction of the MLA monomer [26]. The first part is calculated from the lowest exothermic temperature peak (peak1) is initiated by carboxy oxygen, and the *E_a_* is approximately 110 kJ/mol; the middle one (peak2) is initiated by amide nitrogen, and the *E_a_* is approximately 155 kJ/mol; the highest one (peak3) is caused through radical cyclization, and the *E_a_* is approximately 166 kJ/mol, which is similar to the PAN homopolymer. The above results further prove that the comonomer MLA can initiate the cyclization reaction through the ionic mechanism, promoting the stabilization of P (AN-*co*-MLA) with a lower *Ea*.

## 4. Conclusions

In summary, we synthesized a high molecular weight acrylonitrile copolymer as a carbon fiber precursor by mixing a solvent polymerization of DMSO/deionized water = 60/40 (wt/wt), which is 3.68 times higher than that prepared by the polymerization solution prepared under the same reaction conditions. Both the conversion of the polymerization and molecular weight gradually decreased when the amount of MLA increased in the feed because of the large molecular volume of MLA, and the MLA feed ratio was better when it was less than 3% for actual industrial production. MLA was able to improve stabilization considerably trough two different ionic mechanisms that demonstrated lowered initiation temperature, broadened exothermic peaks, and smaller *Ea*, confirming that the resultant P (AN-*co*-MLA) is a potential precursor for a high-performance carbon fiber.

## Data Availability

The authors declare that all the data in the article are true and valid. If you need to quote, please indicate the source.

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
