# Peer review of "Preparation and Stabilization of High Molecular Weight Poly (acrylonitrile-co-2-methylenesuccinamic acid) for Carbon Fiber Precursor"

_polymers, 2021, doi:10.3390/polym13223862_

Round 1

Reviewer 1 Report

1-The interaction force between carbon fiber and polymer is not clear in this paper.

2-The validation of this paper should be justified bu other published works.

3-The results section is poor without any deep discussion.

4-In figures, it is better to show the phase velocity versus wave number.

Author Response

Dear reviewer,

      Thanks very much for your hard work and valuable suggestions, please see the attachment, thank you !

     Best Regards!

Reviewer 2 Report

In this work, the copolymerization of acrylonitrile with derivatives of itaconic acid in the presence of AIBN. The monomers' reactivity was investigated by using calculation methods. Moreover, the authors estimated the stability of prepared copolymers to avoid the cyclization of acrylonitrile moieties. Different copolymers with various ratios of acrylonitrile to derivatives of itaconic acid were synthesized. However, to improve the quality of the manuscript, the authors should perform some new experiments or add some new phrases:

  • Line 69, page 2: For such a high molecular weight of copolymers, please change g/mol for kg/mol
  • Line 88, page 2: It should be pH not PH
  • The scheme of the polymerization and synthesis of the monomer should be added to the manuscript. The author should state the purity of the prepared monomer, for example, GC/MS or NMR.
  • The authors should add the structure of the monomers and copolymers.
  • The author should describe the synthesis of the copolymer in more detail (amount of added ingredients in g and mol, the order of added monomer, etc.)
  • In which solvent, the author dissolved the polymer and copolymers for viscosity analysis? The viscosity data should be added to the manuscript.
  • The author state the conversion of the monomer. Please explain which method was used to determine it and add the appropriate data.
  • The authors investigated the molecular weight by using the viscosity method. However, this method does not allow for the estimation of the dispersity of the polymer. The author should perform the GPS of the copolymers to confirm the dispersity and uniform character of copolymers.
  • The author state about the structure of the copolymers is only based on the calculation method. In my opinion, the authors should confirm the structures of copolymers by using the NMR method described, for example, in Polymer Chemistry, 12 (2021), 2551-2562.
  • 10) The author state that the addition of MLA stabilized the PAN copolymers and prevent cyclization. This observation will be corrected if the authors obtain the copolymers with -A-B-A-B- structure. In my opinion, MLA does not stabilize the PAN polymer. The less amount of the PAN affected decrease of CN signal in FTIR and DSC or XRD. The authors state that the reactivity of the MLA monomer is higher than AN. It means that the authors obtained the block copolymer, not alternative copolymers. The structure of the copolymers should be confirmed by the NMR technique.
  • To compare the stability effect of MLA comonomer all analyses (DSC, FTIR, XRD, after and before heating) should be performed for PAN homopolymer.
  • According to Figure 10, a low temperature for the cyclization is required for the MLA/AN polymers. Does it mean that the homopolymer is more stable than copolymers?
  • The number of pages and volumes are omitted in references 16 and 28.

Author Response

Dear reviewer,

      Thanks for your hard work and valuable suggestions, please see the attachment ,thank you !

      Best regards!

Round 2

Reviewer 1 Report

Good/Accept

Author Response

Thanks for you hard and careful work again!

Best Regards

Reviewer 2 Report

The manuscript was corrected according to the reviewer's suggestion.
However, one point should be more discussed. 

The authors showed the SEC results according to the reviewer's suggestion. However, the molecular mass in the table is incorrect. It should be 105, not 10-5. The author should be added units for example kg/mol. Additionally, the authors showed very narrow polydispersity of copolymers which is mainly observed for living polymerization. The authors should show the GPC curves, should discuss and explain why they were obtained in classical radical polymerization such narrow dispersity.

Author Response

      Thanks for you hard and careful work again, please see the attachment, thank you !

Best Regards,
